# Phenomic Microglia Diversity as a Druggable Target in the Hippocampus in Neurodegenerative Diseases

**DOI:** 10.3390/ijms241813668

**Published:** 2023-09-05

**Authors:** Daniele Lana, Giada Magni, Elisa Landucci, Gary L. Wenk, Domenico Edoardo Pellegrini-Giampietro, Maria Grazia Giovannini

**Affiliations:** 1Department of Health Sciences, Section of Clinical Pharmacology and Oncology, University of Florence, Viale Pieraccini 6, 50139 Florence, Italy; elisa.landucci@unifi.it (E.L.); domenico.pellegrini@unifi.it (D.E.P.-G.); mariagrazia.giovannini@unifi.it (M.G.G.); 2Institute of Applied Physics “Nello Carrara”, National Research Council (IFAC-CNR), Via Madonna del Piano 10, 50019 Florence, Italy; g.magni@ifac.cnr.it; 3Department of Psychology, The Ohio State University, Columbus, OH 43210, USA; wenk.6@osu.edu

**Keywords:** inflammaging, LPS inflammation, Alzheimer’s disease, ischemia, microbiota, α7nACh receptor, reactive microglia, cannabidiol, TRPV2 channels

## Abstract

Phenomics, the complexity of microglia phenotypes and their related functions compels the continuous study of microglia in disease animal models to find druggable targets for neurodegenerative disorders. Activation of microglia was long considered detrimental for neuron survival, but more recently it has become apparent that the real scenario of microglia morphofunctional diversity is far more complex. In this review, we discuss the recent literature on the alterations in microglia phenomics in the hippocampus of animal models of normal brain aging, acute neuroinflammation, ischemia, and neurodegenerative disorders, such as AD. Microglia undergo phenomic changes consisting of transcriptional, functional, and morphological changes that transform them into cells with different properties and functions. The classical subdivision of microglia into M1 and M2, two different, all-or-nothing states is too simplistic, and does not correspond to the variety of phenotypes recently discovered in the brain. We will discuss the phenomic modifications of microglia focusing not only on the differences in microglia reactivity in the diverse models of neurodegenerative disorders, but also among different areas of the brain. For instance, in contiguous and highly interconnected regions of the rat hippocampus, microglia show a differential, finely regulated, and region-specific reactivity, demonstrating that microglia responses are not uniform, but vary significantly from area to area in response to insults. It is of great interest to verify whether the differences in microglia reactivity may explain the differential susceptibility of different brain areas to insults, and particularly the higher sensitivity of CA1 pyramidal neurons to inflammatory stimuli. Understanding the spatiotemporal heterogeneity of microglia phenomics in health and disease is of paramount importance to find new druggable targets for the development of novel microglia-targeted therapies in different CNS disorders. This will allow interventions in three different ways: (i) by suppressing the pro-inflammatory properties of microglia to limit the deleterious effect of their activation; (ii) by modulating microglia phenotypic change to favor anti-inflammatory properties; (iii) by influencing microglia priming early in the disease process.

## 1. Introduction

For over a century, the brain was seen as a network of neurons, dendrites, axons, and synapses in a space embedded by other cells which, like glue, filled the empty space. Nevertheless, proper recruitment, activation, and intercommunication among neurons and glia is of fundamental importance for the functional organization of the brain. The research is now focused on understanding how microglia engage in morphological, ultrastructural, transcriptional, proteomic, and epigenetic switches that influence their functions, their responses, and their effects on the surrounding cells, suggesting that microglia states are modulated by local cues (for a comprehensive review, see [1]). In this review we shall use the nomenclature indicated by Paolicelli et al. [1].

While for most of the last century “activation” of microglia was considered solely detrimental for neuron survival, it was later demonstrated that microglia are fundamental in early brain development (for ref., see [2]), for synaptogenesis, synaptic maintenance, maturation and synaptic pruning (for ref., see [3]), and in normal learning and memory in mice [4,5,6,7].

Microglia (5–10% of brain cells in number) are myeloid cells that invade the brain early during development [8,9] and coordinate the interactions between the immune system and cognitive functions [3,10,11,12,13,14,15,16,17,18]. Early studies in adult mice demonstrate that microglia acquire diverse morphologies in different brain areas, from radially orientated arborized cells in the grey matter to longitudinally branched elongated cells in the white matter and compact amoeboid cells around the circumventricular organs [19]. For many years, ramified microglia (Figure 1A) were considered quiescent or ‘resting’, while they are now considered “homeostatic” [1]. 

In healthy conditions, microglia have small soma and fine, highly mobile ramified branches, which dynamically reorganize their shape and length, continuously elongating and withdrawing to patrol a defined, non-overlapping territory of the brain parenchyma [20] to detect and eliminate damaged neurons and maintain a healthy environment [21,22,23,24]. Recent studies have described hyper-ramified microglia (Figure 1D) in the medial prefrontal cortex of rats in response to chronic stress [25,26].

The ramified branches of microglia work as chemotactic sensors, moving their extensions towards injured cells in the for phagocytosis [27], and the impairment of their mobility can be deleterious in many conditions. Decreased migration of microglia hampers their phagocytic efficacy, increasing the degeneration of neurons and accumulation of toxic debris [28], and weakening microglia neuroprotective effects. Activation of microglia is a quick process that leads to morphological, phenotypic, and functional changes that stimulate the migration of microglia to the damaged brain area. Reactive microglia, after having fulfilled their phagocytotic function, regress rapidly to the homeostatic form [24]. Furthermore, during apoptotic clearance, spherical phagocytic pouches (ball) are formed at the tip of microglial terminal branches (chain), the so-called ball-and-chain structures (Figure 1C), which can phagocytose apoptotic debris [29] or a small quantity of other substances [29], with no modification of ramified morphology, in contrast to the phagocytosis performed by amoeboid microglia in pathological conditions (Figure 1E) [12]. In mouse organotypic hippocampal slices, ramified microglia exert neuroprotective effects during NMDA-induced excitotoxicity [30].

Microglia are plastic cells, and an oversimplified, already surpassed view, recognized two functional phenomic extremes acquired in response to cytokines, chemokines, and other soluble factors, namely the classical M1 pro-inflammatory and the M2 anti-inflammatory phenotypes [12,27,31,32,33,34,35]. M1 cells were classified as reactive cells that release pro-inflammatory cytokines, such as TNFα, IL-1, IL-6, and IL-18 [36], and have harmful properties. M2 were classified as non-reactive cells that secrete anti-inflammatory cytokines, such as IL-4, IL-10, IL-13, and TNF-ß, and have beneficial, neuroprotective properties, [36].

Nevertheless, the classification of microglia in these two all-or-nothing states [37] is too simplistic and does not correspond to the variety of different microglia phenotypes recently discovered [38,39]. Between these two extreme functional states, a plethora of phenotypically diverse intermediates with different functional states is now recognized. Indeed, a recent hypothesis based on accumulating data postulates that microglia, as neurons, exist physiologically as heterogeneous, mixed populations, which differ in their transcriptomic and morphofunctional characteristics. Consequently, microglia may exist in n possible phenomic states, diverse in health and disease conditions, and which depend not only on the type and intensity of insult and on the progression of the disease, but also on the brain structure where microglia are located [37,40,41,42,43,44]. The vast array of receptors expressed by microglia constantly surveying their surroundings constitutes a ‘sensome’, allowing detection and response to different stimuli that derive from sensory and behavioral experiences [1,45,46].

**Figure 1 ijms-24-13668-f001:**
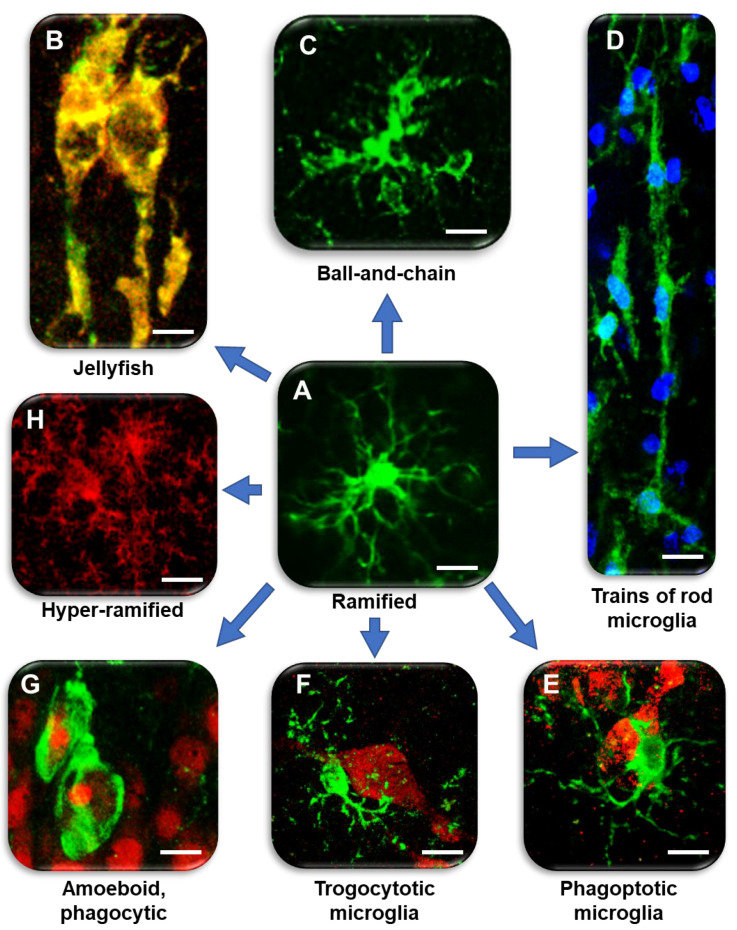
Diversity of microglia morphologies in the hippocampus. (**A**) Ramified, homeostatic microglia. IBA1: green (modified from [47]). Bar: 5 µm. (**B**) Jellyfish microglia in the ischemic CA1. IBA1: green, major histocompatibility complex type II (MHC II): red; merge: yellow–orange (modified from [48]). Bar: 2.5 µm. (**C**) Ball-and-chain-like structures at the end of microglia branches that can phagocytose small amounts of material. IBA1: green (modified from [49]). Bar: 5 µm. (**D**) Train of rod microglia in response to ischemia. IBA1: green; nuclei: blue (modified from [48]). Bar: 10 µm. (**E**) Phagoptotic microglia. IBA1: green; NeuN: red (modified from [50]). Bar: 5 µm. (**F**) Trogocytotic microglia. IBA1: green; NeuN: red (modified from [50]). Bar: 5 µm. (**G**) Amoeboid microglia with rounded morphology and phagocytosing pyknotic neurons after ischemia. IBA1: green; NeuN: red (modified from [48]). Bar: 10 µm. (**H**) Hyper-ramified microglia in CA1 of TGCRND8 mice. CD68: red (modified from [49]). Bar: 5 µm.

Microglia are anything but static, and, since they are exceptionally responsive to alterations in the surrounding environment, the states of microglia vary as a continuum rather than an all-or-nothing phenomenon. It will be interesting to understand whether microglia located in different brain areas acquire identical phenomics [51,52], or whether they react differently to the same insult. For instance, in the CA1 and CA3 hippocampus, microglia reactivity states are different in aging and acute inflammation [47,53], or after an excitotoxic insult [30]. Indeed, Vinet and colleagues [30] demonstrated that ramified microglia exert neuroprotective roles in pathologic processes, and that this function is region-specific, at least in the hippocampal areas. Region-specific differences in lysosome content and membrane properties of microglia [38,54], as well as the expression of genes related to the phagocytic capacity of microglia, have recently been demonstrated [55]. Furthermore, using scRNA-seq, it has been demonstrated that microglia populations display higher transcriptomic diversity in the developing, aged, and diseased brain [56,57] than in the adult brain. Many different receptors with a variety of functions are expressed by microglia. Their expression, and the outcome of their activation, depend not only on the pathological conditions, but also on the functional state of the cell. Depending on the nature of the ligand and on the receptor, downstream intracellular pathways translate their activation with detrimental or beneficial effects (for references, see [58]). The dysfunction of microglia has been described in many CNS disorders, such as AD [59], frontotemporal dementia [60,61], and PD [62].

The complexity of microglia phenomics and their related functions compels the continuous study of microglia in animal models of diseases. In this review, we will discuss how microglia phenotypes change in the hippocampus of animal models of normal brain aging, acute brain inflammation, and neurodegenerative diseases. This emerging, understudied area of research will teach us much about normal and abnormal brain function and will allow to find new druggable targets that will ultimately enable us to intervene in three different ways: (i) by suppressing the pro-inflammatory properties of microglia to limit their deleterious effects; (ii) by modulating microglial phenotypic change to favor anti-inflammatory states; (iii) by influencing microglial priming early in the disease process.

## 2. Different Microglia Phenomics in Inflammaging

The chronic, sterile inflammation that develops during normal brain aging, defined as inflammaging [63], is characterized by slow, low-grade upregulation of pro-inflammatory responses [64,65], and activates microglia that first cooperate to maintain brain homeostasis. If the inflammation becomes more intense, microglia increase the expression and release of pro-inflammatory proteins and neurotoxic mediators [37], which cause the progressive, irreversible changes in the aging brain, leading to many functional changes, such as memory deficits.

Aging, associated with functional alterations in microglia phenotype [66], which become intensely pro-inflammatory [67], is a risk factor for vascular cognitive impairment and dementia (VCID). The phenotypic modifications seem to be caused by alterations in the metabolic pathways of microglia, since an inflammatory stimulus reduces mitochondrial oxidative phosphorylation in microglia [68]. Minocycline, a potent inhibitor of inflammation, has efficacy in preclinical models of VCID, particularly in conditions in which microglia are activated, such as cerebral hypoperfusion [69,70]. Minocycline is protective after chronic administration, possibly reducing the number of microglia cells, and restoring the white matter functions impaired by hypoperfusion [69,70].

It is now becoming clear that changes in microglia phenotypes, rather than inflammation, per se, are among the causes of neuronal degeneration. Expression of upregulated genes in microglia during inflammaging differs from those expressed during different types of acute inflammation [71], demonstrating that reactivity of microglia is a multistage, reversible, highly regulated process that generates many different phenomics with protective capacity [12,27,31], and is more than solely a negative process.

Inflammaging is particularly important in the CA1 and CA3 hippocampal areas, which cooperate with different roles in encoding mechanisms of short-term memory. The comparison between microglia phenomic changes in CA1 and CA3 can give insights into the higher sensitivity of CA1 to neurodegenerative insults, both in experimental animal models and in humans [72,73,74]. In CA1 of aged rats, microglia are less numerous [47], are irregularly distributed, and have reduced cytoplasmic projections [47,53,65,75]. Cytoskeleton remodeling of microglia can be monitored using immunostaining for IBA1, a microglia marker of Ca^2+^-dependent actin polymerization, which decreases during aging [76], indicating that microglia have low efficient cytoskeletal remodeling and, consequently, impaired mobility and a lower patrolling capacity of the brain parenchyma [77]. All these inflammaging-related modifications can hamper the phagocytic activity of microglia, weakening their neuroprotective efficacy and favoring the accumulation of degenerating neurons and pro-inflammatory neuronal toxic debris [28], typical of brain aging [47]. Although the signaling pathways related to phagosome, lysosome, or antigen presentation may be upregulated in inflammaging, less mobile and less numerous microglia can still actively phagocytose and scavenge apoptotic neurons or debris [21,47]. Thus, microglia actively maintain their protective role during normal aging [27,78,79,80] by disposing of dying cells [81]. Membrane-bound fractalkine (CX3CL1) is an adhesion molecule that is expressed on CA1 pyramidal neurons actively phagocytosed by microglia [47]. The cleaved, soluble form of CX3CL1, binding to its receptor (CX3CR1) expressed on microglia, regulates microglia phagocytic activity [82,83,84,85], recruiting microglia towards injured neurons [82,86,87,88]. Microglia [89] and, to a lesser extent, astrocytes [90,91,92] express CX3CR1, and soluble CX3CL1 recruits microglia and astrocytes in a well-organized reciprocal interaction around apoptotic neurons, forming organized neuron–astrocyte–microglia triads in which astrocytes cooperate with microglia to phagocytose dying neurons [47]. Indeed, soluble CX3CL1 increases in cerebral ischemia [93] in response to apoptosis [94] and glutamate stimulation [86] and is neuroprotective in cultured rat hippocampal neurons [95]. In the spinal cord, CX3CL1 contributes to the development of neuropathic pain [96], indicating that CX3CL1’s final effects depend on the localization of the cell and/or on the different stimulus [97].

During normal aging, microglia actively maintain their protective role [27,78,79,80] by clearing dying neurons [81]. However, in a pro-inflammatory context their scavenging activity is considerably decreased, and microglia actively participate in maintaining inflammatory responses that may lead to neurodegeneration [98]. In normal aging, microglia upregulate immune response signaling receptors [99], neuroprotective signaling pathways [100], and pro-inflammatory major histocompatibility complex type II (MHC II), as well as IL-1α, IL-1β, IL-6, TNFα, and C1q [101], which contain lipofuscin granules [102]. It is still not completely understood whether all these age-induced modifications are protective or damaging. The scavenging activity of microglia differs in different conditions, such as aging and acute inflammation [76,102], and in a mouse model of AD the microglia phagocytic activity and clearance capacity were age-related and inversely correlated with deposition of Aβ plaques [103]. In aged animals, IBA1 immunostaining is significantly lower than in animals after acute inflammation with LPS. In addition, in aged rats, microglia are characterized by extremely limited branching, while in LPS-treated rats microglia cells are highly ramified [76]. Amplified, exaggerated, or chronic microglia reactivity, as in chronic inflammatory diseases, can cause robust pathological changes, neurodegeneration, and neurobehavioral complications [67,98].

Apoptosis is a physiological mechanism that helps maintain normal tissue homeostasis through the resolution of low-grade inflammation [104]. In the chronic, low-grade inflammatory conditions typical of normal brain aging, microglia patrol the brain parenchyma to dispose of neuronal debris by phagocytosis, or of entire neurons by phagoptosis (Figure 1G; [97]). Therefore, microglia, reducing the diffusion of inflammatory molecules from apoptotic neurons and neuronal debris, have a protective role in the control of inflammation, thus, preventing further damage to healthy, neighboring neurons.

## 3. Different Microglia Phenomics in the Hippocampus in Acute Inflammation

The ability of microglia to maintain their protective role by clearing dying neurons [81] decreases considerably in a pro-inflammatory context [21]. For instance, in a transgenic mouse model of AD (APP-SL70 mice), microglia phagocytic activity is inversely correlated to Aβ plaque deposition and aging [105].

In acute inflammation, upregulated genes highly increase the expression of NF-κB signaling factors. The quick response of microglia to signals after acute pro-inflammatory damage is possibly due to the rapid activation of NF-kB, which translocates from the cytoplasm to the nucleus. IL-1 and IL-6, pro-inflammatory cytokines, and activators of NF-kB recruit microglia [106]. Interferon-γ and IL-4, released by T cells, stimulate the expression of MHC II on microglia and accelerate microglia proliferation, while matrix metalloproteinases (MMPs), released from apoptotic cells, stimulate microglia reactivity. Reactive microglia rapidly express many receptors of the immunoglobulin family, complement receptors, cytokines, chemokines (IFN-γ, IFN-β, IFN-α, IL-1, IL6, IL-12), and receptors for mannose, acquiring the capacity to recognize and bind various antigens and present them to T lymphocytes [107]. On the other hand, TGF-β1 and IL-10 reduce MHC II expression and negatively modulate microglia.

In acute cerebral inflammation, microglia patrol the parenchyma to prevent the spreading of pro-inflammatory factors from apoptotic neurons and debris [47,53] and to restore normal physiological conditions. A further example of phagocytically active microglia is trogocytosis (Figure 1F), a process first demonstrated in lymphocytes B and T and more recently in resident microglia, which can partially remove small portions of neuronal axons during postnatal development in mice [108]. This process could also be responsible for the removal of small portions of neurons to reduce neuron-dependent damage (see an example in Figure 1F).

Hyper-ramified microglia projections (MPJs) establish numerous contacts with astrocyte branches (APJs) [76], and at the contact sites MPJs/APJs, microglia accumulate the mechano-sensor Integrin-β1 [76,109,110], which possibly influences the dynamic remodeling of MPJs [76] and promote microglia branch extension and/or retraction [111]. Increased expression of integrin-β1 in MPJs may target microglia towards damaged neurons for phagocytosis [76]. Microglia located near damaged, clasmatodendrotic astrocytes have low levels of integrin-β1 in the MPJs, which are shorter, enlarged, and less extended/retracted [76]. As a consequence, these microglia are less mobile. Hyper-ramified microglia found in the medial prefrontal cortex of rats in response to chronic stress [25,26] show upregulation of integrin-β1, and this effect is rescued after the administration of minocycline, a microglia inhibitor.

Under acute inflammatory conditions, astrocytes and microglia secrete plasminogen activator inhibitor type 1 (PAI-1), which has an important role in microglia migration [112] via the low-density lipoprotein receptor-related protein (LRP)-1/Janus kinase (JAK)/STAT1 axis and phagocytic activity via vitronectin and Toll-like receptor 2/6 [113].

## 4. Different Microglia Phenomics in the Hippocampus in Ischemia

After ischemia, the dual roles of microglia may depend on their phenomic polarization [32,33,34,35], since many pro- or anti-inflammatory cytokines and chemokines are produced by microglia after ischemia [114]. During the early phase of vascular damage, the microglia response is protective in the mouse spinal cord [115]. When the damage is prolonged, microglia are inappropriately stimulated into a phagocytic phenotype to remove the dying endothelial cells [116]. In a mouse model of cerebral ischemia, the protective role of microglia is demonstrated by the inverse correlation between proliferation of microglia and extension of the lesion and neurodegeneration [102]. Apolipoprotein E (APOE), regulating triggering receptor expressed on myeloid cells 2 (TREM2), which modulates the activation of microglia [117], is responsible for microglia role in neurodegeneration.

The resting status of microglia and their responsiveness to insults also depend on brain regional differences. In the mouse brain, microglia sensitivity to insults, and their transcriptome and phenomics, vary in a region-dependent way [99]. After ischemia, ATP binds to different P2 purinergic receptors that may have beneficial or detrimental effects [118] in the resolution or aggravation of inflammation [119]. Within the metabotropic P2Y G-protein-coupled receptors, both P2Y2R and P2Y6R promote the phagocytic clearance of apoptotic cells, contributing to the termination of inflammation [120]. On the contrary, microglia P2Y12R seems to be involved in mediating the damage in cerebral ischemia [121]. The P2X7R likely participates in the cerebral damage associated to stroke [122].

Furthermore, it has been demonstrated that the increased number of microglia cells and their reactivity after hypoperfusion [70,123] increase the release of MMP-2 [124], which degrades the extracellular matrix (ECM) and the tight junctions between endothelial cells, causing damage to the blood–brain barrier (BBB) [125]. Increased ROS production by reactive microglia may alter NO signaling, with consequent endothelial dysfunction [126]. Furthermore, MMPs can degrade myelin [127].

The mechanism of apoptosis has a homeostatic function, able to regulate cell death and maintain the number of cells in physiological and pathological conditions [128]. However, it is still uncertain how apoptotic neurons are disposed of. The principal mechanism is probably releasing intercellular signals, such as the find-me signals ATP and CX3CL1 [47,88], and the eat-me signals, such as phosphatidylserine (PS) (reviewed in [129]), which recall and activate phagocytic cells to engulf and consume the neuron. The release of do-not-eat-me signals from neurons, such as CD47-SIRPα or CD200-CD200L, maintains microglia in a quiescent state and suppresses phagocytosis (reviewed in [130]). Microglia express pattern recognition receptors (PRR), such as CX3CR1, P2Y6, P2Y12, stabilin 1, SIRPα, TREM2, MerTK, and CD11b [131,132,133,134], that allow them to recognize molecules released by damaged neurons [86,88], causing the phagocytosis of degenerating neurons and neuronal debris. It is conceivable that the modifications to the microglia phenomics during their activation (such as downregulation of P2RY12 and other receptors) can contribute to pathological dysfunctions of neuron excitability and, consequently, behavioral alterations typical of neurodegenerative disorders, such as AD [135,136,137,138].

First described in 1899 by F. Nissl [139], rod microglia cells were long forgotten and “re-discovered” only after 2012 by Ziebell and colleagues in a rat model of traumatic brain injury [140,141,142], raising interest on their still unknown functions in health and disease [140,141,143,144,145,146]. Elongated bodies of rod microglia align end-to-end, forming trains adjacent to apical dendrites and perpendicular to the dural surface (Figure 1D) [141,147,148]. In 2015, it was demonstrated that trains of rod microglia are present in the CA1 and CA2/3 hippocampus of patients with AD, dementia with Lewy bodies, and hippocampal sclerosis of aging [149], and in a rat model of global ischemia 10 days after the insult [150]. 

Early after an ischemic insult, a significant percentage of microglia form trains of rod cells with elongated and narrow soma, with a retraction of planar processes that span from the SR throughout the entire thickness of CA1 SP, reaching the outer layer of SP where they acquire a round-shaped amoeboid phagocytotic head (Figure 2A, modified from [48]). On the contrary, in CA3 SP, an area more resistant to ischemia [151], no rod microglia trains are found. The different phenomics of microglia in CA1 and CA3 can possibly explain the different sensitivity of the two areas to the same ischemic insult. Rod microglia express MHC II (immunodetected by the antibody OX6, blue in Figure 2A,B; modified from [48]). At longer times after a strong ischemic insult, trains of rod microglia are no longer detectable in CA1, but most microglia cells acquire an amoeboid, phagocytic morphology (Figure 2(C–C2), modified from [48]). Although rod microglia morphology suggests specific functions in pathological states, very little is known on their role in ischemia or other neurological pathologies.

The trains are oriented with their longest axis perpendicular to CA1 SP and are aligned parallel to and in close contact with the apical dendrites of CA1 pyramidal neurons [48]. A similar spatial disposition of trains, perpendicular to the pial surface, is found in models of TBI [140], in human viral encephalitis, HIV-1, and in the general paralysis of the insane [152], as also found by Nissl, who described that the trains were aligned with injured axons [139,153,154,155]. Rod microglia originate by differentiation of existing resident CNS microglia [146,148,156] and migrate in response to chemokine signaling released by damaged neurons [157,158]. Thus, alignment of rod microglia with apical dendrites of CA1 pyramidal neurons may help migration of microglia to the areas with higher injury, to find and phagocytose damaged neurons [159]. Other hypotheses [141] are that rod microglia form trains adjacent to neurons to (i) remove damaged neuronal processes, (ii) form a protective barrier for healthy neurons from a damaging environment, or (iii) isolate a damaged neuron from interaction with healthy neighboring neurons. Nevertheless, the question is still open.

Thus, it appears that in response to some damaging stimuli, quick response of microglia leads to phenomic changes that promote their elongation, train formation, and migration to the damaged areas [20,23,24], where they become phagocytic and eliminate the damaged neurons by phagocytosis. In the outer layer of CA1 SP at early times after ischemia most of rod microglia have enlarged heads and phagocytose pyknotic neurons (Figure 2A,B, modified from [48]), confirming that rod microglia are highly phagocytic cells [141,154,160,161] and in agreement with data that rod microglia are immunoreactive for the phagocytic marker ED1 [140]. The morphological transformation of the rod microglia heads resembles jellyfish microglia (Figure 1B, modified from [48,162]), structures directly related to ATP-mediated microglia activation through the P2RY12 pathway. If the damage to the tissue is not too intense, after fulfilling their functions, phagocytic microglia return to their resting state and continue patrolling the tissue [24].

The time course of rod microglia emergence and resolution is poorly understood, but according to Graeber and Mehraein [152] rod microglia can only form when the tissue is at least partly preserved and not completely damaged after the insult. As mentioned above, after a very strong ischemic insult, when neuronal necrosis is extensive, the scenario completely changes, and in a few more hours most of rod microglia modify their morphology to an amoeboid phagocytic phenotype [152] (Figure 2(C–C2)), modified from [48]). In a mCAO rat model of transient ischemia, rod microglia are formed in the penumbra area of the ipsilateral cortex from 10 min to at least 48 h after the insult [163]. In hippocampal slices exposed to an ischemic insult, microglia in CA1 and CA3 respond differently, indicating that their phenotypic changes are independent from the insult, but respond to internal, spatial-dependent cues, as also found in other brain areas [37,40,41,42,43,44,47,53]. Although it is still not completely understood how and why these two areas of the hippocampus are differently affected by the same ischemic event [151,164,165,166,167,168,169], new data start to shed some light onto the possible mechanisms. One hypothesis is that these two regions differ in the size of their extracellular space. Smaller extracellular space would concentrate cellular debris, have a lower pH, and have higher CO_2_ levels, which could consequently cause differential reactivity of microglia. The modifications that ischemia brings about not only in neurons but also in microglia, and their interrelationships, can be envisaged as one of the causes of the different responses to the same insult. Neuronal vulnerability is paralleled by rod microglia formation in CA1 but not in CA3, where most of pyramidal neurons are spared from the damage, suggesting that rod microglia do not have a protective role towards an ischemic insult, at least in the CA1 hippocampus.

All these data taken together demonstrate that, in response to a damaging stimulus, microglia can acquire different spatiotemporal functional phenotypes, at least in the hippocampus.

## 5. Microglia Phenomics in Alzheimer’s Disease

Data from animal models of AD show that microglia are recruited at the site of Aβ deposition, and regulate Aβ levels in the brain [49], contributing to Aβ clearance and removal of cytotoxic debris from the brain [27,170,171,172,173]. Plaque-associated microglia inhibit additional fibrillization of Aβ and plaque growth [174], thus, protecting neighboring neurons [175]. However, the phagocytic activity and clearance capacity of microglia inversely correlate with Aβ plaque deposition and aging [105].

Microglia responses in AD are influenced by APOE and TREM2 [117]. TREM2 regulates microglia energetic and biosynthetic metabolism [176], maintaining the high activity microglia need to dispose of excess Aβ. However, the intense TREM2-dependent activation of microglia may in turn cause a harmful chronic inflammatory response of TREM2 [117]. Sustained activation of microglia can also cause phagoptosis of healthy neurons [171,177,178,179] intensifying neurodegeneration [65,180], and can cause phagocytosis of synapses in response to soluble Aβ [181], while microglia depletion prevents loss of neurons and dendritic spines [182], further suggesting a pathogenic role for microglia hyperactivation in AD. Variants of TREM2 impair microglia activation, phagocytic properties, inflammatory responses, energy metabolism, and plaque compaction, affecting the progression of AD [176,183,184]. Toll-like receptor 4 (TLR4) in microglia plays an important role in neuroinflammation [185], but studies on TLR4- and on TREM2-deficient mice give conflicting results on AD pathology [186]. In TREM2 deficient mice, loss of microglia clustering around Aβ plaques increases AD risk, supporting the idea that microglia are protective [175]. Nevertheless, it has also been shown that microglia associated with Aβ plaques have a neurodegenerative phenomic, regulated by the TREM2-APOE pathway, which suppresses the phagocytosis of apoptotic neurons [117]. TLR4, stimulated by both fibrillar and oligomeric forms of Aβ [187], seems to be protective [188]. Further, stimulation of Toll-like receptor 2 (TLR2) by fibrillar Aβ activates microglia into a more pro-inflammatory profile, with detrimental effects on AD pathology [187].

The age-related impairment of chemotactic sensors may weaken the neuroprotective activity of microglia which phagocytose Aβ fibrils less efficiently in old than in young mice [66]. On the contrary, amplified, exaggerated, or chronic microglia activation can lead to robust pathological changes and neurobehavioral complications, such as in chronic inflammatory diseases [39,98].

In both AD transgenic mice and human AD brains, a unique subtype of protective microglia, named disease-associated microglia (DAM), has recently been found [39]. DAM contribute to disease mitigation by expressing many factors that enhance microglia phagocytic activity [39], which do not represent the primary cause of the disease, per se, but do affect AD time-course and progression rate. The functions of the genes expressed by DAM are first TREM2-independent and later TREM2-dependent [39], and the transition to fully-activated DAM does not occur in the absence of TREM2 receptors. Increased expression of TREM2 during this stage is a defensive factor linked to Aβ clearance [39,189], indicating that TREM2 is necessary to support phagocytosis at a late stage of the disease. In more advanced stages of the disease, TREM2-expressing microglia, interacting with accumulating neurofibrillary tangles, cause extensive inflammation and neurodegeneration [190,191], but the absence of TREM2 in microglia at this stage of AD, but not at earlier stages, exacerbates AD symptomatology [192,193]. Complement C3, responsible for excessive release of pro-inflammatory mediators and induction of reactive astrocytes [37], is one of the most highly upregulated genes involved in this microglia reaction [194]. Distinct microglia subpopulations, located in different brain areas, seem to have different roles at different times in disease progression [39].

Dark microglia, immunonegative for the homeostatic microglial marker P2RY12 and weakly positive for CX3CR1 and IBA1, were identified from their condensed cytoplasm and nucleoplasm, and for their dilated endoplasmic reticulum and altered mitochondria, which make them distinguishable from typical microglia. Dark microglia interact with blood vessels, axon terminals, and dystrophic dendritic spines, and are highly immunoreactive for CD11b and TREM2 [195]. Dark microglia are found near fibrillar Aβ, near Aβ plaques and dystrophic neurites in the CA1 stratum lacunosum moleculare of the ventral hippocampus of the transgenic strain APP/PS1 [196,197]. Nevertheless, the exact role of dark microglia in the pathogenesis of AD remains unclear [195].

In the hippocampus, microglia have a high “immune-vigilant” phenotype that can be responsible for their higher activation in response to Aβ plaque formation, giving rise to a harmful chronic inflammatory response [183]. Furthermore, hippocampal microglia display lower expression of many proteins, including CXCR3 [185], one receptor involved in neuron–microglia communication, and in microglia recruitment, neuronal reorganization [186], and microglia activation during demyelination [187], as described above. Therefore, decreased levels of the CXCR3 receptor and other proteins in AD-vulnerable brain regions may impair microglia response and recruitment.

Microglia express α7nAChR [198] that possibly drive the “cholinergic anti-inflammatory pathway” that regulates systemic inflammatory responses [199]. In CA1 and CA3 of TGCRND8 mice, a transgenic model of AD, microglia surrounding Aβ plaques (plaque activated microglia, PAM) have a different phenotype and show differential expression of α7nAChR (Figure 3). In TGCRND8 mice, microglia in CA1 are bigger, more reactive, and express higher levels of α7nAChR than in WT mice and in CA3 of Tg mice (Figure 3). In CA3, microglia show a ramified state and express α7nAChR (Figure 3B), while in CA1 microglia have a round cell body with shorter branching, and the expression of α7nAChR is more intense (Figure 3C). This differential response of microglia in CA1 and CA3 around plaques confirms one more time the differential spatial states of the cells. Hart and colleagues [200] showed a further regional difference between microglia located in the white matter versus microglia located in the grey matter [184]. Region-specific variations in gene expression (both increases and decreases) may be implicated in the progression or in the resolution of neurodegenerative diseases [201].

In the hippocampus of AD patients, the subregional pattern of atrophy is different from other groups of neurodegenerative conditions [203,204], and understanding the molecular and cellular mechanisms that lead to such a subregional vulnerability could unveil therapeutic strategies to alleviate the progression of memory decline. Genome-wide association studies (GWAS) identified AD onset risk loci that are associated with genes involved in microglia physiology and responses, such as CR1 (complement receptor type 1), SPI1 (transcription factor PU.1), TREM2 (triggering receptor expressed on myeloid cells 2), and CD33 [205].

Boosting microglia defensive capabilities with cell-specific therapies may offer new avenues for preventing or reversing neurodegeneration. Further work is needed to demonstrate and dissect these features.

## 6. Dysbiosis and Microglia

From the recent literature, it appears that microglia respond to gut microbiota composition by integrating multifarious signals and modifying their responses from neuroprotective to neurotoxic effects. How the gut microbiota can control microglia responses remains poorly understood and is a matter of research.

In germ-free (GF) or specific pathogen-free (SPF) mice, the morphology of microglia is severely altered, and the cells have longer, very mobile, hyper-ramified projections (see an example in Figure 1H), which partially overlap in the spatial domains of neighboring microglia [206,207,208] entering into physical contact with the projections of adjacent cells. A possible functional consequence of these alterations of microglia projections is the altered pruning of synapses [209]. Bacterial-produced molecules, such as SCFAs, LPS, peptidoglycans and PAMPs (pathogen-associated molecular patterns) [210], can cross both the IB and the BBB [211], reaching the brain parenchyma where they can be recognized by TLR4 expressed on microglia, playing important roles in neuroinflammation [185].

Dysbiosis causes increased Aβ production, Aβ plaque deposition, and increased activation of microglia that migrate to the sites of Aβ plaques, interact with Aβ deposits, and regulate Aβ levels in the brain [212]. In comparison to APP/PS1 mice with normal microbiota, GF APP/PS1 mice have less Aβ and decreased plaques formation, as well as decreased microglia [213], indicating that signals from the microbiota delineate microglia phenomics, while dysbiosis causes microglia dysfunctionality. Microbiota reintegration partly reverses the microglia cell morphology and functionality [206]. The microbiota seems necessary for the maturation and maintenance of microglia in proper physiological conditions, ready to display a rapid response to damaging stimuli [206].

Many hypotheses have been postulated to explain how the microbiota can regulate microglia, including the following: (I) SCFAs produced by microbiota cross the BBB and in the CNS target microglia, regulating their function or maturation; (II) after interacting with SCFAs, immune cells expressing SCFAs receptors can migrate to the brain through the BBB; (III) other metabolites or compounds produced by the microbiota, called microbe-associated molecular patterns (MAMPs), can cross the BBB, target microglia, and regulate their function or maturation; (IV) peripheral macrophages that recognize MAMPs migrate to the brain and cross the BBB; (V) the gut microbiota can communicate directly with the microglia via the vagus nerve [214] that senses changes in pro-inflammatory cytokines caused by gut inflammation, and through its afferent fibers sends information to the CNS, influencing microglia and inflammatory mechanisms.

Furthermore, epigenetic mechanisms can shape the phenomics of macrophages during development, but the local microenvironment within and outside the brain can additionally reprogram the genetic imprint [206,215]. Little is known about how epigenetic mechanisms can control microglia phenomics. Interestingly, prenatal ablation of histone deacetylases1/2 (HDAC1/2) impairs microglia development, while it has no effect on microglia homeostasis in adult mice [216]. In a mouse model of AD, HDAC1/2 deficiency in microglia increases amyloid phagocytosis, resulting in decreased Aβ load and amelioration of cognitive impairment [216]. It appears, therefore, that epigenetic factors, which can have different outputs during development or in adulthood, affect microglia maturation, homeostasis, and activation in a differential manner. The gut microbiota can affect epigenetic modifications throughout the entire lifespan, as has been demonstrated in diabetes and obesity [217], but possibly also in AD.

## 7. Spatiotemporal Differences in Microglia States in the Hippocampal Areas

Understanding the differences in microglia phenomic states, spatial distribution, and reactivity may help explain the differential susceptibility of the hippocampal areas to aging or to different inflammatory insults [46,218]. Microglia show marked hippocampal dorsoventral, interregional, and interlaminar differences [219,220], and show unique morphological, ultrastructural, and physiological features between the two hippocampal poles. Microglia depletion by PLX, their modulation by minocycline, or the absence of the Cx3cr1 gene strongly affect LTP in a region-specific manner, indicating that proper signaling between microglia and neurons is crucial for the maintenance of physiological plasticity along the longitudinal axis of the hippocampus [46]. Remarkably, in the adult hippocampus, microglia density is lower in CA1 than in CA3 and, during aging, microglia further decrease in CA1 but increase in CA3 [47,53]. The expression of genes encoding markers of phagocytic activity and mRNA of Mertk and CD68 are upregulated in the ventral but not in the dorsal hippocampus. Mertk coordinates astrocyte–microglial crosstalk for the phagocytosis of dead neurons [221].

Reactive microglia are seen diffusely scattered throughout the brain after 2 days of LPS infusion [222], but during the following weeks reactive microglia gradually decrease in all cerebral regions, and after 4 weeks the greatest inflammatory response is concentrated within the hippocampus [222]. Thus, LPS initiates a cascade of biochemical processes that cause time-dependent, regional, and cell-specific changes [222,223], consistent with the hypothesis that microglia recognize danger signals in the parenchyma, including those released by cellular debris produced from apoptotic cells, and cooperate and help in clearing apoptotic neurons or neuronal debris [224,225]. This concerted action can prevent or reduce the release of pro-inflammatory mediators and subsequent injury to neighboring neurons [226,227].

Furthermore, it should not be forgotten that direct cell–cell interactions among neurons, astrocytes, microglia, and oligodendrocytes can influence and mediate microglia branching, causing branch extension towards pro-inflammatory triggers in an orchestrated manner. The disruption of cell–cell interactions could be responsible for the dysregulation of microglia defensive activity [76]. However, the intracellular pathways that regulate phenomic changes during surveillance as well as the nature of the interactions between microglia and neighboring cells remain unclear. Dissing-Olesen and colleagues [228] showed that microglia can drastically change their morphology in response to environmental cues, and extracellular ATP and ADP are the main extension factors in the process [111]. A unique phenotype characterized by bulbous structures extending at the end of microglia processes in response to ATP injections and neuronal damage [23] was found. Microglia acquire unique morphofunctional phenotypes, and the classical subdivisions between “branched meaning resting” and “amoeboid meaning phagocytic” do not correspond to the actual multiple and complex phenotypes of microglia, since many studies have described that microglia are phagocytically active even in the branched state [16,29], confirmed by other studies in mice [229] and zebrafish. The formation of the bulbous endings is transient and is reversed when ATP application is discontinued [228,229]. These results suggest that ATP plays an essential role in signaling between neurons and microglia, that microglia can detect neuronal activity, and that this activity can be modulated through bulbous microglial processes.

## 8. Microglia as a Druggable Target

The different spatiotemporal molecular mechanisms involved in the modification of microglia phenomics of reactivity and polarization so far described are complex, diverse, and not completely understood. The different morphofunctional phenotypes of microglia found in CA1 and CA3 areas of the hippocampus in different models of neurodegeneration shows the complexity of microglia differentiation even just in this relatively small part of the brain (Figure 1). Each of these morphofunctional phenomics have been described and can be considered as druggable targets, possibly responding to or being the cause of differential outcomes of drug therapies. Treatment strategies so far, which function by generally interfering with activated microglia receptors, signal pathways, and inflammatory mechanisms, are not fully effective. For instance, classical FANS or coxibs (e.g., aspirin, celecoxib, and naproxen) failed to prevent or treat neurodegenerative diseases [230], demonstrating that inhibition of pro-inflammatory microglia alone is not sufficient to have a positive outcome and that concomitant activation of anti-inflammatory microglia may be needed.

Therefore, understanding the spatiotemporal heterogeneity of microglia phenomics is crucial for the development of novel microglia-targeted therapies in different CNS disorders. Some steps have been taken in the last few years, but the turning point to find specific and effective therapeutic strategies will be the understanding of the complex, differential responses of microglia to different insults. A few examples are described below.

Microglia homeostatic molecules, such as CX3CR1, P2Y12, and TGF-b1R, are downregulated in disease-associated microglia [117,231,232,233,234], and modulation of these molecules could be an important druggable target to restore microglia defensive mechanisms during CNS insults.

In a mouse model of cerebral ischemia, P2X7 receptor antagonists attenuate the production of pro-inflammatory factors in the hippocampus and improve memory deficits and animal survival [235], and TLR4 gene mutation blocks the activation and release of pro-inflammatory factors by microglia in AD [236].

Microglia express endocannabinoid receptors [237] and it has been hypothesized that cannabinoid 2 receptor (CB2R) signaling shifts the balance from neuroinflammatory to neuroprotective and homeostatic gene expression, by which cannabinoids acting on microglia acquire therapeutic functionality [237,238,239]. The cannabinoid compound cannabidiol (CBD) modulates many genes involved in the regulation of inflammation [240], thus, restraining microglia-mediated inflammatory responses [241,242,243,244]. In addition, CBD, acting on receptors other than the CB2R, enhances microglia phagocytosis through activation of transient receptor potential (TRP) channel [245]. Expression of the TRPV2 channel increases after ischemia in reactive phagocytic microglia [238], playing a key role in phagocytosis [246]. CBD enhances TRPV2 expression and promotes its translocation to the cell surface [246], successfully attenuating neuroinflammation while simultaneously improving mitochondrial function and ATP production via TRPV2 activation [245]. TRPV2 channels seem to be part of a feed-forward amplification of intracellular Ca^2+^ signaling that enhances microglia reactivity [246]. Nevertheless, it should be taken into consideration that the protective effect of CBD on microglia may be mediated by receptors other than the TRPV2 channel.

Nicotine can significantly decrease the levels of TNF-α and IL-1β in the hippocampal CA1 region of rats after ischemia by upregulating α7nAChR and inhibiting microglia inflammation and JAK2/STAT3 [247]. Microglia express α7nAChRs [198] that modulate a cholinergic pathway which regulates microglia activation through α7nAChR, since nicotine prevents LPS-induced TNF-α release [198]. Activation of microglia α7nAChRs by nicotine [248] increases the expression of cyclooxygenase-2 and PGE2 synthesis while the antagonist MLA, or the partial agonist GTS-21, unveil an anti-inflammatory role of α7nAChR [249]. This microglia-driven signaling seems crucial for neuroprotection. In organotypic cultures subjected to an ischemic insult, microglia α7nAChR has protective effects [250]. After global ischemia in the rat, the beneficial, anti-inflammatory effect of nicotine is associated with the proliferation of microglia. The microglia α7nAChR is also involved in phagocytosis [251], a phenomenon possibly mediated by cytoskeleton reorganization and increased release of cytosolic Ca^2+^ from intracellular stores [252]. In human microglia, nicotine induces the expression of many genes, such as TGF-β1, IL-4, CX3CL1, CCR2, and CXCR6 [253], and in models of inflammation nicotine has an anti-inflammatory effect reducing TNF-α and IL-1β release [254,255]. It should be pointed out that chronic exposure of cells to α7nAChR agonists causes upregulation of the receptor, possibly secondary to agonist-dependent receptor desensitization [256]. The upregulation of the receptor might explain the neuroprotective effects obtained with repeated administration of the agonists.

CD33 expression by microglia is inversely related to Aβ uptake capacity [257], indicating that increased CD33 expression in microglia promotes plaque pathology [257]. Therefore, CD33 seems a regulator of microglial Aβ uptake, and its inhibition could increase microglia Aβ uptake capacity, which in turn would be beneficial in AD [258]. Inhibitors of CD33 are hypothesized to increase phagocytic properties in microglia.

Voltage-independent KCa3.1-like Ca^2+^-activated K^+^ [259,260] are expressed by microglia. KCa3.1 is expressed on CD68-positive microglia after an ischemic insult [261]. The study by Chen et al. demonstrated the therapeutic efficacy of inhibiting KCa3.1 with TRAM-34 in ischemic injury. The KCa3.1 inhibitor TRAM-34 reduces LPS-induced microglial neurotoxicity [262], and inhibition of KCa3.1 using senicapoc blocks the LPS-induced IL-1β and NO production by microglia [263]. Senicapoc is a drug closely related to TRAM-34, and its safety has been established in clinical trials, in which it was tested as a treatment for asthma and sickle-cell anemia. The ability to “re-purpose” senicapoc for the treatment of Alzheimer’s disease may help it more rapidly advance to clinical trials [264].

Other studies have since demonstrated the therapeutic efficacy of KCa3.1 inhibition in other models of neurodegenerative disease in which microglia are believed to play a significant role [265,266].

## 9. Conclusions

The microglia activation profile is not an all-or-nothing phenomenon, but rather a continuum of different levels of activation states, which depend on the type of insult and its progression, as well as the areas of the brain where the microglia cells are located [38,54]. Indeed, microglia are continuously active, surveying the parenchyma and reacting to external damaging stimuli in a different manner, depending on the stage of life, brain region, species, and sex, by adopting different states and performing different functions [1] (see Table 1).

Future studies should be aimed at further unravelling our understanding of inter-glia and intercellular communication in the CNS, such as the microglia-mediated activation of astrocytes and neuronal degeneration. Identifying the molecules and signaling pathways that trigger such intercellular responses or intracellular morphophysiological transformations will undoubtedly help us to understand the cues leading to microglial differentiation (morphologically and functionally).

Understanding the heterogeneity of microglial phenotypes in different neuropathology is key to resolving the microglia-mediated neuroinflammation associated with various CNS disorders. Identifying the molecules and signaling pathways that trigger such intercellular responses or intracellular morphophysiological transformations will undoubtedly help us to understand the cues leading to microglia differentiation (morphologically and functionally).

In this review, we have pointed out the diverse morphological and functional states that microglia can undertake in the hippocampus in aging and neurodegenerative disease, and how microglia’s reduced ability to rapidly respond to brain challenges can lead to CNS dysfunction and disease progression. Both intrinsic and extrinsic signals collaborate to dynamically determine microglia functional states and, consequently, point to relevant functional differences in health and disease. Therefore, understanding the spatiotemporal heterogeneity of microglia phenomics is of paramount importance for the development of novel microglia-targeted therapies in different CNS disorders.

## Figures and Tables

**Figure 2 ijms-24-13668-f002:**
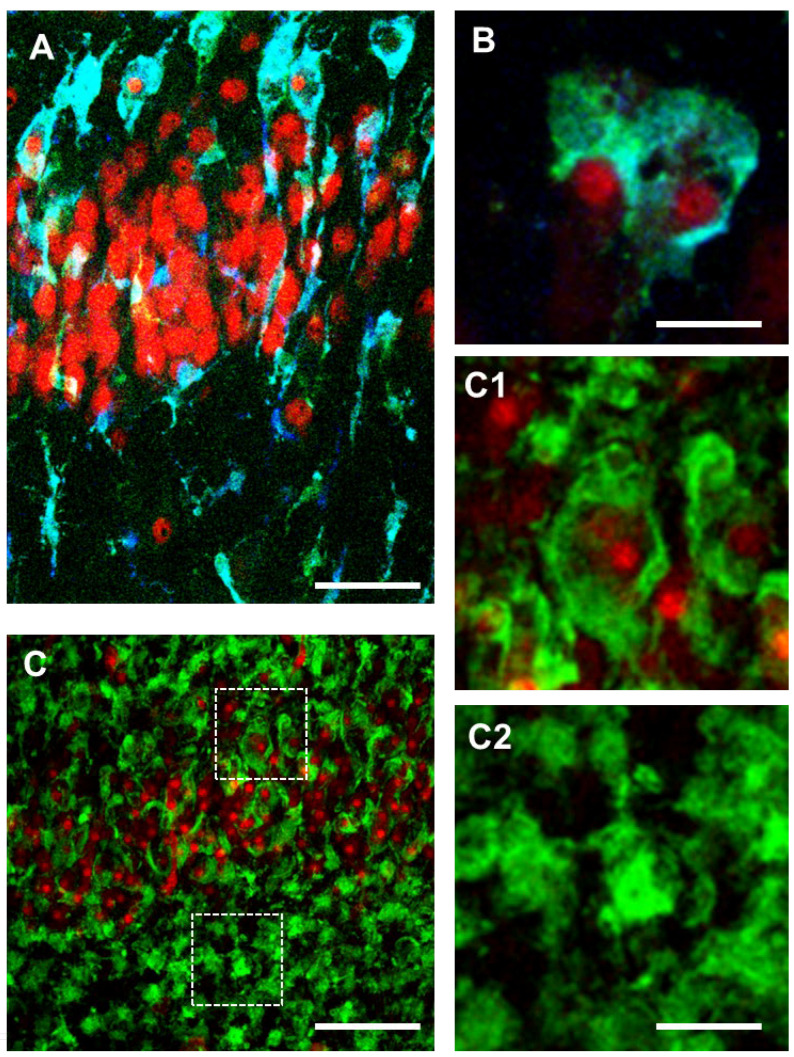
Representation of the different stages of microglia activation, in CA1 of organotypic hippocampal slices 24 h (**A**,**B**) and 36 h (**C**–**C2**) after OGD. Anti IBA1 antibody (green), anti MHC II antibody (blue), anti NeuN antibody (red). Colocalization of IBA1 (green) and MHC II (blue) is evidenced by the cyan colour (**A**,**B**). Scale bars: 25 µm (**A**); 10 µm (**B**,**C1**,**C2**); 50 µm (**C**) (modified from [48]).

**Figure 3 ijms-24-13668-f003:**
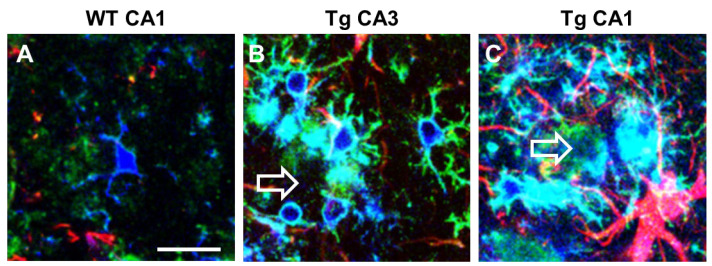
Intense upregulation of α7AChNR in reactive plaque-associated microglia in CA1 and CA3 of TgCRND8 mice. Comparison with WT mice. Anti IBA1 antibody (blue), anti α7AChNR antibody (green), anti GFAP antibody (red). Empty arrows indicate Aβ plaques. (**A**) A microglia cell with faint α7AChNR immunofluorescence in CA1 hippocampus of WT mouse. (**B**) Ramified plaque-associated microglia with intense α7AChNR immunofluorescence in CA3 of a TgCRND8 mouse. (**C**) Amoeboid plaque-associated microglia with very intense α7AChNR immunofluorescence in CA1 of a TgCRND8 mouse. Astrocytes are also visible in red. Scale bar: 20 µm (from [202]).

**Table 1 ijms-24-13668-t001:** Spatiotemporal heterogeneity of microglia phenomics in CA1 hippocampus in different models of neurodegeneration.

Microglia Phenotype	CA1 SP	CA1 SR/SLM	References
Jellyfish	+		[48,162]
Ball-and-chain		+	[29]
Trains of rod microglia	+	+	[48,140,142,147,149,150,156]
Phagoptotic		+	[65,97,171,177,178,179,180]
Trogocytotic		+	[108]
Amoeboid phagocytic	+	+	[141,142,152,154,159,160,161]
Hyper-ramified		+	[206,207,208]
DAM		+	[39]
Plaque-associated microglia		+	[105,117,174,175,176,183,184,187,188,212]
Dark microglia		+	[195,196,197]

## Data Availability

All authors of this review agree to share the unpublished research data upon request.

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
