# Peer review of "Phenomic Microglia Diversity as a Druggable Target in the Hippocampus in Neurodegenerative Diseases"

_ijms, 2023, doi:10.3390/ijms241813668_

Round 1

Reviewer 1 Report

1.       Proofreading for minor English grammatical corrections.

2.       Line 127: “Furthermore, using scRNA-seq it has been demonstrated that microglia population display higher transcriptomic diversity in the developing, aged, and diseased brain [53]” refer to PMID: 35714196. 

3.       Line187: “both microglia [85] and astrocytes [86] express CX3CR1”. There are multiple reports on astrocytes having a very low baseline level expression of CX3CR1. Please correct the statement accordingly.

4.       Many of the statements appear repetitively in multiple sections. For example, “Microglia produce a plethora of pro- or anti-inflammatory cytokines and chemokines” appear under multiple sections. Please make them concise.

5.       The sections describing the role of microglia in diverse types of hippocampal insults should be grouped together under one section and subsections for easy understanding.

Minor english editing is required.

Author Response

We thank the referee for the favourable review given to our manuscript.

  1. Proofreading for minor English grammatical corrections.

We have gone through the entire text and made some corrections, as suggested. We do hope that the text is now suitable for publication.

  1. Line 127: “Furthermore, using scRNA-seq it has been demonstrated that microglia population display higher transcriptomic diversity in the developing, aged, and diseased brain [53]”

The reference Singh N, et al., 2022. (PMID: 35714196) has been added, as requested.

  1. Line187: “both microglia [85] and astrocytes [86] express CX3CR1”, There are multiple reports on astrocytes having a very low baseline level expression of CX3CR1. Please correct the statement accordingly.

The statement has been corrected and 2 references added (Hatori  et al., 2002; Sheridan and  Murphy, 2013)

  1. Many of the statements appear repetitively in multiple sections. For example, “Microglia produce a plethora of pro- or anti-inflammatory cytokines and chemokines” appear under multiple sections. Please make them concise.

The idea of our review is to underline that microglia responses are different in different physio/pathological contexts and we believe that some repetitions are necessary. Nevertheless, we tried to make them as concise as possible.

  1. The sections describing the role of microglia in diverse types of hippocampal insults should be grouped together under one section and subsections for easy understanding.

We are sorry to disagree with the reviewer. Indeed, the organization of the review is based on the idea to subdivide the phenotypes and responses of microglia in CA1 and CA3 hippocampus following the different insults, rather than grouping them.

Reviewer 2 Report

The review article submitted by Daniele Lana et al. entitled “Phenomic Microglia Diversity as a Druggable Target in the 2 Hippocampus in Neurodegenerative Disease” discusses the role of microglia in the brain and their types and functions in different areas of the brain. This review provided the most updated progress in microglia-related research and described most of the technologies used in this research field. Overall, this is a decent review article and will benefit readers who are interested in microglia research work. More importantly, it highlighted the potential application of the phenotypes and location differences of microglia for drug targeting. This manuscript is written in precise professional language. However, it will be even more clear if the author can make a table or a diagraph to describe the microglia distribution and microglia types and functions along with the references.

Author Response

We thank the referee for the very positive comments to our review.

As suggested, we have added a table “Spatiotemporal heterogeneity of microglia phenomics in CA1 hippocampus in different models of neurodegeneration” with the relevant references.